# Enhancing Nursing Excellence: Exploring the Relationship between Nurse Deployment and Performance

**DOI:** 10.3390/ijerph21101309

**Published:** 2024-09-30

**Authors:** Reni Asmara Ariga, Rebecca Aurelia, Paskah Thio Dora Anak Ampun, Cindy Patresia Hutabarat, Ferdinand Batiscta Panjaitan

**Affiliations:** Department of Fundamental Nursing, Universitas Sumatera Utara, Medan 20155, North Sumatra, Indonesia; rebecca.aurel@gmail.com (R.A.); paskahthiodoraanakampun@gmail.com (P.T.D.A.A.); cindypatresiahtb02@gmail.com (C.P.H.); ferdinandpjt@students.usu.ac.id (F.B.P.)

**Keywords:** placement, executive nurse, performance, nursing excellence, nursing care

## Abstract

Proper nurse placement is crucial for enhancing the performance and quality of health services. This study aims to explore in-depth the relationship between nurse placement and performance in order to promote nursing excellence. A quantitative analysis was conducted using a descriptive correlational methodology. The population in this study consisted of 214 executive nurses at Medan Government Hospital, with a sample size of 139. The study’s findings revealed that nurses performed exceptionally well in providing nursing care, scoring 94.2%, with those well-placed scoring 90.6%. The results from the Spearman rho correlation test showed that nursing qualifications, experience, work environments, and team dynamics have significant relationships with nurse performance. Meanwhile, the nurse’s rho factor towards patients and the policy or regulation component showed low significance and relationship. The novelty of this study lies in its indication that nursing performance can be enhanced by aligning the placement of nurses with their abilities and experience, and fostering a work environment and positive team dynamics that encourage collaboration and efficiency. These findings provide vital insights for nursing staff management in order to enhance nursing care quality and patient health outcomes. This study highlights the need for suitable placement and professional development for nurses in order to attain maximum performance.

## 1. Introduction

In the ever-changing world of healthcare, nurses play a critical role in providing excellent treatment to patients. As demands and complexity in the global health system rise, nurse placement emerges as a critical element influencing nurse performance and the quality of care offered to patients. This phenomenon is increasingly relevant with the increasing need for nursing staff throughout the world, which demands a deep understanding of the influence of nurse placement on the final outcome of care. The World Health Organization (WHO) states that 59% of health workers are nurses, with around 28 million worldwide. The WHO estimates that by 2030 there will be a need for 36 million nurses worldwide to meet public health needs [1]. Although the growth of the nursing workforce varies significantly across countries and regions, the shortage of nurses and the problem of distribution of the proportion of the nursing workforce continue to pose challenges in health services worldwide [2].

In Indonesia, there are 147,264 nurses working in hospitals, representing 45.65% of all medical personnel working in hospitals. In 2014, Indonesia’s nurse-to-population ratio was 94.07 per 100,000 people, but this decreased to 87.65 in 2015. This proportion falls short of the 2014 aim of 158 nurses per 100,000 people, as well as the target of 180 nurses per 100,000 population stated in the Indonesian Ministry of Health’s strategic plan for 2015–2019. These statistics highlight the uneven distribution of nursing professionals in Indonesia, as well as the inadequate number of nursing workers [3].

Human Resource management (HR) is crucial for attaining an organization’s goals. This is particularly true in hospitals, where nurse placement is vital for developing high-performing nursing teams and promoting nursing excellence [4]. Nurses play a significant role in coordinating healthcare efforts that promote public health. Furthermore, the role of placing nurses is closely related to organizational operations, such as aiding individuals in meeting organizational goals. Nurse managers deploy senior nursing experts to each nursing unit in order to provide the best care and patient comfort [5].

Many factors impact the management of nurses’ employment in hospitals, including experience and credentials, nurse-to-patient ratios, policy or legal requirements, the job’s environmental conditions, and dynamics within teams. These criteria are meant to ensure that nurses are optimally positioned to enhance care quality, patient health outcomes, nursing excellence, and performance. The location of nurses in care settings has an impact on patient health [6]. According to empirical evidence, effective placement and a sufficient number of nurses prevent negative outcomes for patients receiving nursing care [7].

Nurse performance refers to the delivery of nursing care to patients based on nurse professionalism, as well as all associated activities and processes [8]. Nursing care impacts the overall quality of care received, and having an adequate number of nurses decreases adverse occurrences and improves outcomes [9]. Nurses in workplaces with inadequate staffing will work ineffectively and inefficiently, for instance, failing to observe vital signs on time or failing to respond to patient needs. This leads to missed care as well as negative patient perceptions of the quality of care [10].

Nurse placement with nurse performance has a significant impact on the needs and challenges in today’s global healthcare. Appropriate nurse placement is critical to addressing health workforce shortages, especially in light of an aging population, increased case complexity, and rising demand for healthcare services. Effective nurse placement ensures that nurses are not overwhelmed with their assigned task load. In addition, judicious nurse placement can impact nurses’ physical and mental wellbeing by providing a welcoming work environment, reducing stress, and preventing burnout [11].

Optimal deployment of nurses can improve the quality of patient care through better supervision, better case management, and standardized care [12]. This results in overall health system efficiency, including reduced costs, shorter patient wait times, and more effective use of resources.

Research on nurse placement also has an impact on patient safety. The placement of nurses whose expertise and experience match what is required to handle complicated cases improves their ability to provide appropriate and quality care to ensure consistent care for patient safety. Effective nurse placement can also improve the quality of care, reduce the risk of errors, and improve patient safety during the care process [13,14]. Effective deployment can also help organize nursing teams, ensure good collaboration between nurses and other healthcare professionals, and improve care coordination and patient safety [15].

To address the shortage of nurses, research on the relationship between nurse placement and nurse performance is essential. This research can identify key factors that affect performance, such as nurses’ qualifications and experience [16], nurse-to-patient ratios, policies or regulations, work environments, and team dynamics in the institution or unit.

This study aims to recommend optimal placement models for health institutions to maximize nurses’ potential, improve the quality of health services, and create a work environment that supports the growth and sustainability of the nursing profession.

Previous research has highlighted the relationship between nurse deployment and care efficiency, but deeper understanding of the factors influencing this relationship is needed. Although there is a consistent correlation between nurse assignment and patient outcomes, the mechanisms governing nurse assignment still require comprehensive elucidation [17]. By advancing knowledge in this area, this research intends to provide valuable insights for health practitioners, researchers, and policymakers, in an effort to raise the global standard of nursing care in response to the demands of an ever-changing and developing world. The aim of this study is to explore the correlation between nurse placement and performance, and to show how this relationship affects the quality of nursing care provided to patients within the current healthcare context.

## 2. Materials and Methods

### 2.1. Design and Participants

This was a quantitative research study using a correlational descriptive approach and a cross-sectional study design. It was conducted in a class-A public hospital that has facilities and medical service capabilities of at least 4 (four) Basic Specialist Medical Services, 5 (five) Specialist Medical Support Services, 12 (twelve) Other Specialist Medical Services and 13 (thirteen) Sub-Specialist Medical Services. This type of public hospital is not owned by all regions in Indonesia and has the function of being a referral hospital for lower types of hospitals. The population of this study consisted of 214 and the research samples are 139 executive nurses at the Medan Government Hospital during 2024. Data collection was conducted from May 3 to July 9 2024. The sample approach utilized was the probability method, namely, simple random sampling in a probability sample, each unit having the same probability or chance of being selected [18]. Inclusion criteria were executive nurses who worked in inpatient units A and B as well as executive nurses who were willing to be respondents.

### 2.2. Study Procedures

Stage 1 (Figure 1) consisted of collecting literature reviews related to the research topic. Literature sources were obtained through PubMed, ScienceDirect, and the Directory of Open Access Journals (DOAJ). The questionnaire in this study consisted of 3 parts: first, a demographic data questionnaire formulated by the researcher; second, a nursing staff placement questionnaire adapted from Cooper’s questionnaire on clinical placement evaluation instruments; and third, a nurse performance instrument adapted from Nursalam’s questionnaire. First, a demographic data questionnaire was distributed to examine the characteristics of respondents consisting of initials, age, gender, highest level of education (the last educational qualification is the respondent’s last formal level of education), employment status (information that explains the respondent’s position in the hospital such as permanent or contract employment status), length of work, and current work unit. Second, the scientific literature was reviewed regarding questionnaires analyzing the placement of nursing staff. Questionnaires related to nursing staffing were adapted from the Cooper questionnaire to establish an accurate clinical placement evaluation instrument. Cooper’s research was conducted among 1263 nursing students across all year levels of Australian Universities in 2019–2020 [19]. Third, review the scientific literature was reviewed related to nurse performance questionnaires. The nurse performance questionnaire, adopted from the nurse performance questionnaire, of an assessment of nurses in providing nursing care which consists of the assessment process, nursing diagnosis, nursing intervention, implementation, and evaluation [20].

The primary data collection procedure was collecting data directly from respondents and the method used was distributing questionnaires. The questionnaire consisted of a number of statements and was typed and printed as many times as there are respondents. Questionnaires distributed to respondents were expected to be understood and answered [21]. The research implementation continued with data input and validation [22]. The data collected from the questionnaire were then entered into Microsoft Excel and analyzed using IBM SPSS 26 software. The criteria for the results were the level of placement of nursing staff and the level of performance of nurses. The test used was the Spearman rho correlation test.

Stage 2, the validation test process, was carried out in 2 stages. The first stage involved the process of testing content validity, namely, measuring the extent to which the items in the instrument cover all aspects of the construct being measured. This was usually carried out by experts in the relevant field. Further validation was carried out with the IBM SPSS 26 software tools to conduct reliability tests. The validity value of the nurse staff placement questionnaire which adapted the Cooper questionnaire was 0.722 and the Cronbach alpha reliability value was 0.834. The adapted questionnaire included 15 inquiries using the Likert scale (1. firmly disagree, 2. disagree, 3. somewhat agree, 4. agree, and 5. firmly agree). The results of measuring the independent variable, namely the placement of nurses, were divided into 5 categories: very good = 67–75, good = 54–66, quite good = 41–53, not good = 28–40, very bad = 15–27 [18]. For the performance measurement questionnaire, we adapted the performance questionnaire based on Nursalam. The validity test results were 0.966 and the Cronbach alpha reliability value was 0.950. The questionnaire consisted of 25 statements with answers using a Likert scale (1. never done, 2. sometimes done, 3. partially done, 4. often done, and 5. always done). The category of the results of measuring the dependent variable, namely nurse performance, were divided into 5 categories: very good = 109–125, good = 88–108, quite good = 67–87, not good = 46–66, very bad = 25–45 [19].

Next, the second stage consisted of carrying out validity tests directly on samples to assess the suitability of the questionnaire to the research population [23]. In this study, questionnaires were distributed to 30 working nurses for analysis, modification, and adjustment to the research location.

At stage 3, the questionnaire, now valid and suitable for use, was distributed to the research sample of 139 working nurses. The data were collected again to be processed, coded, analyzed, and presented as research results.

### 2.3. Data Analysis

The variable analysis carried out in this study implemented IBM SPSS 26 software. The hypotheses in this study were: H0 = There is no significant relationship between nurse deployment and nurse performance, and Ha = There is a significant relationship between nurse deployment and nurse performance. For the univariate analysis, frequency distribution was used, while for the bivariate analysis, Spearman’s rho was employed to test the hypothesis.

### 2.4. Ethical Considerations

This research received approval from the Committee of Ethics for the Implementation of Health Research at the Universitas Sumatera Utara (with letter number 354/KEPK/USU/2024), followed by approval from the administration of the research location, namely the Medan Government Hospital. Informed consent forms were provided to respondents since participation in this research was voluntary, anonymous, and data confidentiality was ensured. Informed consent was obtained before the questionnaire was administered. Respondents filled out the informed consent form before completing the questionnaire. The questionnaire was then collected from each respondent on the same day that it was completed.

## 3. Results

Research results enhancing nursing excellence: exploring the relationship between nurse deployment and performance data obtained characteristics of respondents, frequency and percentage distribution of nurse staffing questionnaire in Appendix A
Table A1 and Appendix B
Table A2.

### 3.1. Demographic Characteristics of Respondents

From the total sample of 139 executive nurses, the results obtained from the characteristics of respondents in this study were that the majority of respondents were female, (112 respondents, or 80.6% of the total) and the majority of respondents were early adults, 26–35 years old (50 respondents, or 36% of the total). The data were sorted in detail, as the following table shows Table 1.

### 3.2. Distribution of Nurse Placements and Nurse Performance in Providing Nursing Care

The results of this research indicate that the majority of the placements of nursing staff at the Medan Government Hospital were rated very good. A total of 126 respondents (90.6%) considered the nursing staff placement facilities to be very appropriate or suitable for future clinical placements. Additionally, the research results showed that the majority of nurses providing nursing care in the inpatient ward at Medan Government Hospital performed very well, with 131 respondents (94.2%) rating their care as very good. The data were sorted in detail, as the following shows Table 2.

### 3.3. Relationship between the Placement of Executive Nurses and the Performance of Nurses in Providing Nursing Care in the Inpatient Room of the Medan Government Hospital

The outcomes of this study verify that there is a relationship between the placement of executive nurses and the performance of nurses delivering nursing care at the Medan Government Hospital. This is indicated by the Spearman rho sig value. (2-tailed) 0.000 where a value ≤ 0.05 indicates a significant relationship, and there is a moderate relationship according to the *p* value = 0.557. The data were sorted in detail, as the following shows Table 3.

### 3.4. Cross Correlation Test of Indicators for Placement of Nurses with Nurse Performance in Providing Nursing Care

The r and *p* values are obtained from the Spearman test. The research results show that of the five placement indicators, three have a significant and moderate correlation. One of them is qualifications and experience, which has a *p* value of 0.000 and r = 0.505, and team dynamics, which has a *p* value of 0.000 and r = 0.537. The work environment indicator has a *p* value of 0.000 with r = 0.431. Indicators of nurse-to-patient ratio and policy or regulation have a *p* value of 0.003 and r = 0.251, with a weak relationship interpretation. The data were sorted in detail, as the following shows Table 4.

## 4. Discussion

Based on research findings, the majority of respondents consider placement to have a significant and very important influence. The importance of an educated and qualified workforce in the context of hospital services cannot be denied; without the presence of a good workforce, hospitals can experience irregularities in services. Manpower management of either nurses or nursing assistants is essential for good hospital services [24]. In addition, unequal availability of nursing staff can have a negative impact on the functioning of the healthcare system as well as patient safety [25].

Nurse placement factors include nurse qualifications and experience, nurse-to-patient ratio, regulatory policies or standards, work environment, and team dynamics [6].

### 4.1. Qualifications and Experience

Nurse performance and nurse placement are two important components in health service management. Appropriate placement of nurses in roles that match their skills, experience, and interests can improve performance, job satisfaction, and the quality of patient care. The results of the nurse placement study showed that the qualifications and experience of nurses were highly significant and moderately correlated with nurse performance, with a *p* value of 000 and r 505. Most nurses (80.58%) strongly agreed that nursing managers identify nurses’ learning needs, and 79.86% of nurses strongly agreed that nursing managers understand how to assess clinical skills. This suggests that nursing managers should identify nurses’ learning needs and facilitate their access to relevant training. This also influences the placement of nurses according to clinical ability and expertise, which can significantly improve nurses’ performance.

Nurses who are assigned to areas that match their expertise tend to provide more effective and efficient care. Nursing managers who understand nurses’ clinical abilities can ensure proper placement, provide constructive criticism, and support nurses’ professional development. Therefore, managerial involvement in assessing and supporting nurses’ clinical skills is essential. Ultimately, this leads to improved outcomes. Nurse assignments can affect performance in various ways, and specialty nurses with specialized training (e.g., ICU, ER, or pediatric) are better placed in more appropriate settings [26].

Appropriate placement of nurses can increase self-confidence and ability in nursing practice, including the ability to identify patient health problems and apply high-quality standards of nursing care [22]. Nurses with experience and expertise will perform better in comparable settings. For instance, a nurse with extensive oncology experience will be more effective in an oncology ward than in a regular medical ward. Placements that provide growth and development opportunities make nurses more engaged and successful [27].

Placements that offer opportunities for professional development and regular training to improve nurses’ skills and knowledge, as well as appropriate placement based on a thorough evaluation of abilities, interests, and qualifications are essential to achieving optimal nursing performance [28]. Research has shown the connection between emotional intelligence, nursing performance, and years of experience, and has discovered that nurses with higher emotional intelligence performed better and participated more actively in professional development. Moreover, high-performing nurses with situational awareness improved their abilities with experience [29].

### 4.2. Nurse-to-Patient Ratio

The ratio of nurses to patients has a weak correlation with nurse performance, characterized by values of *p* 0.003 and r 0.251. This is in contrast to McHugh’s study, which found that increasing the number of nurses significantly reduced the level of emotional exhaustion [30], Another study showed that hospitals with more dissatisfied nurses also had a lower proportion of dissatisfied patients. However, the nurse-to-patient ratio does not have too much influence on nurse performance if there is adequate management and adequate support from other health teams [31]. Nurse-to-patient ratios vary greatly across the world. Countries in Africa and Southeast Asia have the lowest ratios, while those in Europe have the highest—nearly 10 times greater than those in the lowest regions—at the national level. Numerous factors contribute to these shortages, and an increasing amount of data indicates that the provision of healthcare and its results are impacted by comparatively low headcount. The primary reasons for the nursing shortage are inadequate planning and workforce allocation. Supply-related problems can be addressed in order to find solutions for the nursing shortage. This entails keeping and fostering connections with these extremely uncommon nurses in addition to enhancing recruitment, retention, and return. According to studies, nurses are drawn to the workforce and want to remain there because it allows them to advance their careers, gain independence, take part in decision making, and receive a living wage. For a lasting solution, more demand-side targeted actions are also required. This should be predicated on the understanding that, because healthcare is a labor-intensive service, nursing resources must be used efficiently. As previously stated, there are deficiencies in the health system concerning the quantity and quality of nurses [32].

### 4.3. Policies or Regulations

Policies or regulations have a weak significance, with nurse performance characterized by values of *p* 0.003 and r 0.251 **. In this research, policy or regulation had a low correlation with nurse performance. Nurse staffing policies or regulations are critical, as they have a direct impact on workload balance, quality of patient care, operational efficiency, and safety for both patients and nurses. By establishing appropriate nurse-to-patient ratios, these policies ensure that each patient receives adequate attention, which contributes to improved quality of care and responsiveness to patient needs. Indecisive or incompetent management may result in poor implementation of these policies [33]. A study investigating the experiences of senior nurses and their perceptions of the implementation of the national clinical leadership policy found that they were largely of the opinion that it had a positive impact on team performance, patient care, and clinical leadership. However, it also found that there was a need for a stronger correlation between the management of change and strategic direction [34].

### 4.4. Environment

Research on nurse placement indicates that the work environment significantly and moderately correlates with nurse performance, characterized by values of *p* 0.000 and r 0.431 **. This is indicated by the majority of questionnaire results from 139 (110, or 79.14%) respondents stating that they strongly agree that they felt physically, emotionally, and culturally safe in the work unit environment and that their work placements provided appropriate learning environments.

The experience and abilities of medical staff are not the only factors that influence the effectiveness of health services. Work circumstances are also a significant consideration in understanding how nurses’ work environment affects the quality of treatment and can lead to improvements in the healthcare system [35]. The conditions of a nurse’s workplace are also a factor that influences the placement of a nurse. The clinical environment plays a critical role in the career decision-making process because it prepares nurses to confront the reality and challenges of their work. Nursing decisions made in the clinical setting have a direct influence on the health of patients and the quality of care they receive [36].

This finding is in line with Wareing’s study, which showed that nursing decisions for their first staff nurse employment are heavily influenced by the quality of support offered by mentors and clinical staff, the potential to make a difference in patients’ lives, and the range of available placements [37]. According to the results of the study, most of the nurses had worked for more than 19 years. Nurses have greater confidence and expertise with age and work experience [38]. Research validates this by identifying less experienced nurses with 0–2 years of clinical experience as advanced novices [39].

### 4.5. Team Dynamics

The research results show a significant and moderate correlation between team dynamics and nurse performance, with values of *p* 0.000 and r 0.537 **. This is indicated by the majority of questionnaire results from 139 respondents: 105 (75.54%) strongly agreed that nurses get support to work according to their expertise; 101 (72.66%) strongly agreed that nurses have the opportunity to interact and learn with other health workers; 101 (72.66%) strongly agreed that nurses gain skills and knowledge to develop their practice; and 109 (78.42%) strongly agreed that senior nurses are willing to work with new nurses.

These findings suggest that support and skill development can be factors that strengthen relationships and cooperation within teams, collaboration and an inclusive teamwork culture are important to improving job performance and satisfaction, and orientation to clinical practice and an environment that supports learning can positively influence team interactions and performance.

Further analysis of the research highlights that team dynamics are strongly correlated with the effectiveness of nursing cooperation on unperformed nursing care and nurse-reported care quality. It was found that better coordination, competence, and work productivity were associated with better outcomes [40]. Team learning activities have an impact on the performance of the aged-care nursing team, as knowledge sharing improves performance. Process reflection has an influence on team productivity, efficiency, and innovativeness [41].

Team performance is also positively correlated with simulation-based training sessions that improve communication about key nursing processes such as assessment, planning, implementation, and evaluation [42]. Additionally, social and emotional support from coworkers can reduce fatigue and stress, improve mental health, and encourage nurses to work more. Moreover, cooperation and collaboration in teams allows for more effective problem solving and the division of tasks according to each member’s expertise. Strong team dynamics also encourage mutual learning, where team members share experiences and knowledge and provide mentorship to new members [43]. Finally, a transformational leadership style can assist a leader to ensure that the organization’s mission is carried out efficiently. This leadership can enhance soft skills, job satisfaction, and achieve common goals to provide high-quality nursing care that meets community expectations and safety standards [44].

Based on these findings, the authors suggest the following policies or strategies that can be used to boost performance through nurse placement: (1) increasing nurses’ knowledge and proficiency through ongoing training; (2) increasing team dynamics through effective communication and team-building; (3) improving the work environment by promoting the welfare of nurses; and (4) regularly assessing the nurse-to-patient ratio and revising policies to ensure their applicability and efficient execution in order to raise the standard of nursing services. The novelty of this research lies in its valuable insights for hospital human resource management, with the aim of improving the effectiveness of nurse placement and, ultimately, the quality of healthcare.

## 5. Conclusions

This research had a number of limitations, including the fact that it was limited to one class-A hospital on the island of Sumatra and that it did not cross-test every variable indication. The outcomes of this research confirm that the placement of nurses plays an important role in, and has a significant influence on, their clinical performance in providing nursing care. Appropriate nurse placement based on qualifications, experience, and competencies can improve performance, job satisfaction, and the quality of patient care. In addition, successful nurse placement can improve confidence and practice skills and provide greater opportunities to develop as a nurse. An ideal nurse-to-patient ratio, as well as effective placement policies, help to improve care quality and the wellbeing of nurses. However, competent management and healthcare team assistance are critical in executing these rules. A positive work environment, complete with mentor support and the opportunity to make a difference, influences nurses’ performance and career choices. Strong team dynamics, including effective teamwork, competence, and productivity, as well as social and emotional support from coworkers, have been demonstrated to reduce stress and boost performance. Overall, a nurse’s appropriate placement, a supportive work environment, and favorable team dynamics are essential for reaching peak nursing performance and providing excellent patient care.

## Figures and Tables

**Figure 1 ijerph-21-01309-f001:**
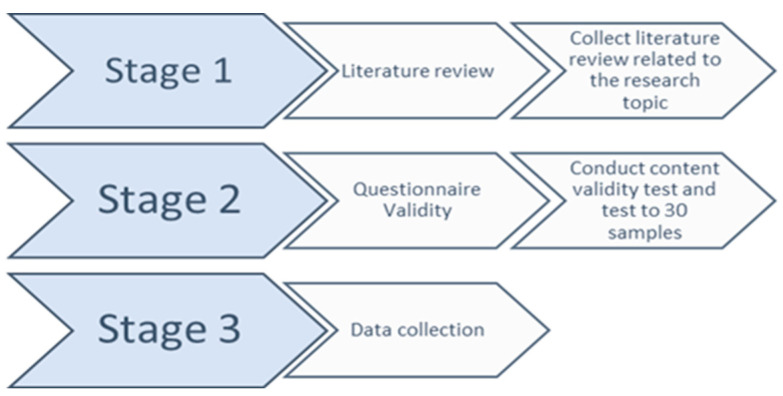
Research stage.

**Table 1 ijerph-21-01309-t001:** Characteristics of respondents (*n* = 139).

Characteristics	*n*	%
Gender		
Female	112	80.6
Male	27	19.4
Age		
Late adolescence		
17–25 Years	5	3.6
Early adulthood		
26–35 Years	50	36.0
Late adulthood		
36–45 Years	37	26.6
Early old age		
46–55 Years	38	27.3
Late old age		
56–65 Years	9	6.5
Last education		
Diploma	67	48.2
Registered nurse	72	51.8
Length of work		
≥1 Years	12	8.6
≥4 Years	43	30.9
≥10 Years	29	20.9
≥19 Years	55	39.6
Employment status		
Still	119	85.6
Contract	20	14.4

**Table 2 ijerph-21-01309-t002:** Distribution of nurse placements and nurse performance in providing nursing care (*n* = 139).

Variable	*n*	%
Placement of Executive Nurses		
Very good	126	90.6
Good	13	9.4
Nurse Performance in Providing Nursing Care		
Very Good	131	94.2
Good	8	5.8

**Table 3 ijerph-21-01309-t003:** Relationship between the placement of executive nurses and nurse performance of nurses in providing nursing care.

Category	Variable				*p* Value	Correlation Coefficient
	Placement		Performance			
	*f*	%	*f*	%		
Very good	126	90.6	131	94.2		
Good	13	9.4	8	5.8	0.000	0.557
TOTAL	139	100	139	100		

**Table 4 ijerph-21-01309-t004:** Relationship between the placement of executive nurses and nurse performance.

Variable	Performance		Performance
	*p*	r	
Qualifications and experience	0.000	0.505 **	The relationship is very significant and moderate
Nurse-to-patient ratio	0.003	0.251 **	Weak relationship
Policy or regulation	0.003	0.251 **	Weak relationship
Work environment	0.000	0.431 **	The relationship is very significant and moderate
Team dynamics	0.000	0.537 **	The relationship is very significant and moderate

** Correlation is significant at the 0.01 level (2-tailed).

## Data Availability

The data presented in this study are available upon request from the corresponding author. Data is not publicly available due to participant privacy.

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
