# Peer review of "Enhancing Nursing Excellence: Exploring the Relationship between Nurse Deployment and Performance"

_ijerph, 2024, doi:10.3390/ijerph21101309_

Round 1

Reviewer 1 Report

Comments and Suggestions for Authors

The paper contains useful information. A number of suggestions could improve the quality of the work in terms of better understanding by readers. In the methods section, clarify what is meant by the answers offered about last educational qualification and employment status. Also briefly describe the second part of the questionnaire, which is the adapted Cooper questionnaire. Under Table 5, put p and r from which test. How are the P-value and the correlation coefficient marked in the tables? If other studies are mentioned in the discussion, add to the first mention where they were carried out and on which sample. Try to present individual answers to the questions contained in the questionnaire in the results.

Author Response

Dear Reviwer 1,

Thank you for the team reviewers who have reviewed the author's article with ID: ijerph-3145376 all the results of the review of reviewers 1 have been reviewed by the author and all input has contributed very well to the perfection of the author's article along with this authors describe the results of the review of revewer 1 and the improvements.

Reviewer 2 Report

Comments and Suggestions for Authors

Dear Authors

It has been a pleasure to have the opportunity to review your manuscript “Enhancing Nursing Excellence: Exploring the Relationship between Nurse Deployment and Performance”.

I would like to make some recommendations to improve the paper:

1.     Introduction

    LINE 75-78. “With an enhanced knowledge of this …”. This paragraph seems like a justification, so it should come before the objective.

2. Methodology

The methodology needs to be significantly improved:

o   LINE 82-83. You state: “The population of this study consisted of 214 executive nurses at the Medan Government Hospital during 2024”. I recommend including the exact period of data collection (e.g. 1 January 2024 to 20 May 2024).

o   LINE 83-84. You indicate: “The sampling approach utilized was the probability method, namely simple random sampling”. This statement must be included in the previous sentence because it deals with the type of sampling, merging it “the population ……2024, the selection was made through a sampling of type …

o   LINE 94. In the statement “The questionnaire used in this research consisted of 3 questionnaires”, this point needs clarification. It may be a questionnaire with three sections or three constructs (1st sociodemographic variables, 2nd placement of nursing staff, 3rd nurse performance), or it may be three independent questionnaires. In any case, you must indicate the origin of the sections/constructs, indicating whether they are self-prepared (1st sociodemographic variables) and the other two, validated questionnaires (cite).

o   Within the methodology, I think it is more appropriate for a better understanding by the reader to include two sections; 1st the procedure (which you already have), and 2nd the measurement instruments. This last point allows us to convey the variables included in each section of the questionnaire clearly, as well as the items that each section has and how this questionnaire is evaluated ranges, interpretation...

o   LINE 106-107. “The primary data collection procedure is collecting data directly from respondents and the method used is distributing questionnaires” It’s essential to clarify how the population was accessed (they went to their workplace, institutional email, etc.). Was consented to participate requested before handing out the questionnaire or, on the contrary, was acceptance considered when filling out the questionnaire? I believe the format of the questionnaire is in paper format. If so, what was the process of questionnaire collection?

o   LINE 135-138. Data analysis you indicate “Variable analysis carried out in this study used IBM SPSS 26 software. For univariate analysis used frequency distribution analysis and for bivariate analysis to test the hypothesis, namely Spearman Rho analysis”. In this subsection I have some doubts; the first being that when I go to the objective of the study I do not find the null hypothesis. On the other hand, it is recommended to remove from “stage 1” and “stage 2” (subsection 2.2 study procedures) the statistical analyses carried out and include them in this point in an organized way.

o   LINE 143. Indicates that informed consent was given. When? Was it together with the questionnaire? When was the completed questionnaire collected? Was consent collected? This should be clarified here or in the procedure section.

2.     Resultados

o   LINE 148-151. They indicate that the final sample was 139 participants. Here I doubt whether the 139 participants were the ones who filled out the questionnaire, even if it was not in its entirety, or are they only the participants who, once the questionnaires were reviewed by the researchers, confirmed that they had provided answers to all the items (an essential criterion to be part of the sample). Clarify the criteria once the researchers receive the questionnaires; this point is also not clear in the procedure – methodology.

o   Tables 2 and 3. Due to the limited data contribution of these tables, it would be advisable to unify them.

o   LINE 195-197. Review the text “This is indicated by the Spearman Rho Sig value. (2-tailed) .000 where a value ≤ 0.05 indicates a significant relationship and there is a strong relationship according to the p value = 0.557”. Review, 0.557 is a moderate but not strong relationship (+/-0.4- 0.59 moderate; +/- 0.6 strong; > +/-0.8 very strong; +/-1 perfect). Also, the text does not match the terms used in the table (eg. P value=0.057 in the text while in the table it is correlation coefficient).

o   Table 5. Add the meaning of ** to the base of the table. Review the strength of the relationships based on the previous comment.

3.     Discusion

o   LINE 242 and 289. Review the document in general to avoid periods before the citation brackets “.[24]” “.[35].”

 Add limitations of the study.

After reviewing, I think it is appropriate to make minor revisions before publication.

Good luck with your article.

Author Response

Dear Reviwer 2,

Thank you for reviewers who have reviewed the author's article with ID: ijerph-3145376 all the results of the review of reviewers 2 have been reviewed by the author and all input has contributed very well to the perfection of the author's article along with this authors describe the results of the review of revewer 2 and the improvements.

Reviewer 3 Report

Comments and Suggestions for Authors

Thanks to the editor for inviting me to participate in the review of this research. This paper discusses the relationship between nurse deployment and performance in enhancing nursing Excellence: exploring the relationship between nurse deployment and performance, aiming at improving nursing Excellence. Although the research provides some valuable insights, there are still some shortcomings that need to be further revised and improved.

Insufficient research significance:

The research background fails to fully explain the urgency and importance of the research. Globally, the shortage of nurses is an increasingly serious problem, but this paper does not explicitly link the research with the current global health care needs and challenges. It is suggested that the author emphasize the importance of nurse allocation to improve nursing quality and patient safety, and its potential contribution to solving the shortage of nurses. It is necessary to highlight the contribution and value of this study.

The summary of existing research is not comprehensive;

The literature review part needs to reflect the breadth and depth of the existing research more comprehensively. In this paper, the references are limited, and the theoretical and empirical research related to nurse allocation and performance can not be comprehensively reviewed. It is suggested that the author expand the literature review, including international perspective and multidisciplinary research, in order to enhance the theoretical basis and practical relevance of the research.

Unreasonable research design:

The research design part lacks detailed description and rationality demonstration of research methods. For example, sample selection seems to be limited to nurses in a regional hospital. It is suggested that the author adopt more diverse samples, which may include different types, sizes and geographical locations of medical institutions, so as to improve the representativeness and extrapolation of the research. Or explain the reasons or limitations of choosing this area. In addition, it is suggested that the potential deviations and limitations of the study should be clarified, and corresponding mitigation measures should be put forward.

The analysis of research results lacks key indicators:

The research results lack in-depth analysis of key performance indicators. For example, nurses' job satisfaction, patient feedback and nursing quality indicators are very important in nursing performance evaluation. It is suggested that the author discuss the relationship between them and nurse allocation in the result analysis. What needs to be explained is the report on some key indicators, which are hardly elaborated by the authors and need to be added.

The discussion lacks pertinence:

The discussion part needs to discuss the significance and practical application of the research results in more detail. The discussion in this paper is broad, and it fails to discuss how to apply the research results to improve the nurse allocation strategy and improve the nursing quality. It is suggested that the author put forward specific policy suggestions and practical guidance according to the research results, and how to customize the nurse allocation strategy according to the specific needs of different medical institutions.

In addition, firstly, this study needs to increase the detailed description and verification process of the research tools to ensure the reliability and effectiveness of the questionnaires and tools used. Second, make clear the innovation of the research and its contribution to the existing knowledge system. Third, strengthen the limitations of research and suggestions for future research direction.

Generally speaking, this paper provides a preliminary understanding of the relationship between nurse allocation and performance, but it needs to be improved in terms of theoretical framework, rigor of methodology, in-depth analysis of results and pertinence of discussion in order to improve the academic contribution and practical application value of the research. It should be pointed out that the format is very bad and needs to be modified.

Author Response

Dear Reviwer 3,

Thank you for the reviewers who have reviewed the author's article with ID: ijerph-3145376 all the results of the review of reviewers 3 have been reviewed by the author and all input has contributed very well to the perfection of the author's article along with this authors describe the results of the review of revewer 3 and the improvements.

Round 2

Reviewer 3 Report

Comments and Suggestions for Authors

The authors made most of the revisions. But I suggest putting the key data in the appendix into the result analysis.

Author Response

Thank you for the team of editors and reviewers who have reviewed the author's article with ID: ijerph-3145376 all the results of the review of reviewers 1,2,3 have been reviewed by the author and all input has contributed very well to the perfection of the author's article along with this we describe the results of the review of revewer 1,2,3 and their improvements.

We have added reviewer #3's input regarding the abstract and background.

Regarding the explanation that appears twice, namely at 4.3 and 4.4, we apologize and we have corrected it at point 4.3
